

# Gαq and Phospholipase Cβ signaling regulate nociceptor sensitivity in *Drosophila melanogaster* larvae

Joshua A. Herman, Adam B. Willits and Andrew Bellemer

Department of Biology, Appalachian State University, Boone, NC, United States of America

## ABSTRACT

*Drosophila melanogaster* larvae detect noxious thermal and mechanical stimuli in their environment using polymodal nociceptor neurons whose dendrites tile the larval body wall. Activation of these nociceptors by potentially tissue-damaging stimuli elicits a stereotyped escape locomotion response. The cellular and molecular mechanisms that regulate nociceptor function are increasingly well understood, but gaps remain in our knowledge of the broad mechanisms that control nociceptor sensitivity. In this study, we use cell-specific knockdown and overexpression to show that nociceptor sensitivity to noxious thermal and mechanical stimuli is correlated with levels of Gαq and phospholipase Cβ signaling. Genetic manipulation of these signaling mechanisms does not result in changes in nociceptor morphology, suggesting that changes in nociceptor function do not arise from changes in nociceptor development, but instead from changes in nociceptor activity. These results demonstrate roles for Gαq and phospholipase Cβ signaling in facilitating the basal sensitivity of the larval nociceptors to noxious thermal and mechanical stimuli and suggest future studies to investigate how these signaling mechanisms may participate in neuromodulation of sensory function.

Corresponding author
Andrew Bellemer,
bellemerac@appstate.edu

## INTRODUCTION

Nociception is the process by which potentially harmful environmental stimuli are detected and processed by the nervous system to generate appropriate physiological and behavioral responses. An understanding of the cellular and molecular mechanisms used in nociception can provide insight into the evolution of sensory systems and cellular pathways that are dysregulated in chronic pain. The larvae of *Drosophila melanogaster* have provided a useful experimental model for understanding nociception. Studies of nociceptive behavior in fly larvae have identified temperature-activated ion channels required for thermal nociception (*Neely et al., 2011*; *Tracey et al., 2003*; *Zhong et al., 2012*), ion channels required for mechanotransduction (*Guo et al., 2014*; *Kim et al., 2012*; *Mauthner et al., 2014*; *Zhong, Hwang & Tracey, 2010*), and receptors required for avoidance of UV light (*Xiang et al., 2010*). Furthermore, the development of a UV-induced tissue-damage model of nociceptive hypersensitization has allowed the identification of evolutionarily conserved signaling pathways that adjust the sensitivity of peripheral nociceptors in response to injury (*Babcock, Landry & Galko, 2009*; *Babcock et al., 2011*; *Im et al., 2015*). Studies of nociceptive

neural circuits have provided insight into how nociceptive information is processed and modulated by other modalities of somatosensory information to shape behavior (*Hu et al., 2017*; *Ohyama et al., 2015*; *Yoshino et al., 2017*). Despite our increasing understanding of larval nociception, we do not fully understand the cellular and molecular mechanisms that determine the basal sensitivity of nociceptors and their downstream circuits.

*Drosophila* larvae respond to noxious stimuli by executing a series of rolls around their long body axis (*Hwang et al., 2007*; *Tracey et al., 2003*). This behavior has been termed nocifensive escape locomotion (NEL) and has been used to identify many of the cellular and molecular mechanisms that control nociceptor function. Class IV multidendritic neurons (mdIVs) are the polymodal nociceptors of *Drosophila* larvae, and manipulations that affect mdIV sensitivity produce predictable changes in NEL. Recent studies have demonstrated that neuromodulatory signaling targets the mdIV neurons to shape nociceptive responses. Following UV-induced tissue damage, tachykinin G protein-coupled receptors (GPCRs) are activated in the mdIV neurons to produce increased electrical excitability and hypersensitivity to thermal stimuli as indicated by faster and more frequent NEL responses (*Im et al., 2015*). The mdIVs also integrate mechanosensory information from non-nociceptive sensory neurons through the activity of modulatory interneurons that release short neuropeptide F (sNPF) onto mdIV axon terminals that express sNPF GPCRs (*Hu et al., 2017*). These neuromodulatory inputs serve to facilitate activation of the mdIV neurons and enhance NEL in response to harsh mechanical stimuli. These findings together indicate that GPCRs and heterotrimeric G protein signaling act as general mechanisms for adjusting the sensitivity of larval nociception.

Signaling through GPCRs on primary nociceptor neurons is a well-understood mechanism for modulating nociceptive sensitivity in mammals. Activation of opioid and cannabinoid GPCRs produces an analgesic effect via inhibition of nociceptor excitability and neurotransmitter release, while activation of serotonin and bradykinin receptors may have pro-algesic effects by increasing nociceptor activation and excitability (for reviews see: *Geppetti et al., 2015*; *Veldhuis et al., 2015*). Bradykinin receptors are notable for their ability to activate nociceptive responses and hyperalgesia via the Gαq-phospholipase Cβ (PLCβ) signaling mechanism, which modulates TRPA1 ion channel activity (*Bandell et al., 2004*; *Bautista et al., 2006*; *Wang et al., 2008*). Part of this modulation arises from the ability of PLCβ to hydrolyze phosphatidylinositol-4,5-bisphosphate (PIP2) into diacylglycerol (DAG) and inositol triphosphate (IP3). In patch-clamp experiments, application of PIP2 to inside-out patches strongly inhibits TRPA1 activity, suggesting that depletion of PIP2 by PLCβ is a mechanism to increase channel activity (*Kim, Cavanaugh & Simkin, 2008*). However, PIP2 may also support TRPA1 channels' recovery from desensitization (*Karashima et al., 2008*). PLCβ signaling also promotes the surface expression of TRPA1 in HEK293 cells to increase channel activity (*Schmidt et al., 2009*).

The regulation of TRPA1 channel activity by Gαq and PLCβ is particularly relevant to *Drosophila* nociception, as the TRPA-family channels, dTRPA1 and Painless, are required in larval nociceptors for thermal, mechanical, and UV-light nociception in *Drosophila* larvae (*Neely et al., 2011*; *Tracey et al., 2003*; *Xiang et al., 2010*; *Zhong et al., 2012*). It is unclear how dTRPA1 and Painless may be regulated by Gαq and PLCβ in *Drosophila*

nociceptors, but dTRPA1 is known to be activated by PLCβ activity in olfactory, gustatory, and thermosensory systems (*Kim et al., 2010*; *Kwon et al., 2010*; *Kwon et al., 2008*; *Shen et al., 2011*). Together, these findings suggest the hypothesis that Gαq and PLCβ signaling may be mechanisms by which the sensitivity of larval nociception is regulated. In this study we address this hypothesis and use cell-specific RNA interference (RNAi) and gene overexpression studies to demonstrate that Gαq and PLCβ signaling regulate the basal sensitivity of *Drosophila* mdIV neurons to thermal and mechanical stimuli.

## MATERIALS & METHODS

### *Drosophila* stocks and genetics

All larvae were reared on cornmeal-molasses medium (Nutri-Fly M; Genesee Scientific, El Cajon, CA, USA). Flies and larvae in all experiments were grown at 25 °C and ∼50% humidity under a 12 h light: 12 h dark cycle. Larvae for all experiments were obtained by washing wandering 3rd instar larvae from the walls of vials using distilled water. The driver stocks *w; ppk1.9-GAL4; UAS-dicer2* and *w; ppk1.9-GAL4, UAS-mCD8::GFP; UAS-dicer2* were obtained as gifts from the laboratory of the Dr. Dan Tracey as was the $w^{1118}$ control stock and the *UAS-para-RNAi* line. All other RNAi lines were developed as part of the *Drosophila* Transgenic RNAi Project (TRiP) and obtained from the Bloomington *Drosophila* Stock Center (BDSC). GL01048 (BDSC# 36820), JF02390 (BDSC# 36775), and JF02464 (BDSC# 33765) were used as *UAS-Gαq-RNAi* lines. JF01713 (BDSC# 31197) and JF01585 (BDSC #31113) were used as *UAS-norpA-RNAi* lines. The $y^1 v^1$; {*Py*[+t7.7]=*CaryPattP*}*2* (BDSC# 36303) line was used in driver-only control crosses for RNAi experiments. *w*\*; {*Pw*[+mC]=*UAS-Galphaq.R*}*2* (BDSC# 30734) flies were used for *UAS-Gαq* overexpression experiments. The $norpA^{36}$ mutant was obtained from BDSC# 9048.

With one exception, the RNAi lines used in this study have been previously validated, as described in the *Drosophila* RNAi Stock Validation & Phenotypes (RSVP) database (*Perkins et al., 2015*). All three *UAS-Gαq-RNAi* lines used in this study have positive validation data for knockdown efficacy, showing embryonic lethality phenotypes following broad knockdown (*Yan et al., 2014*; *Zeng et al., 2015*). One *UAS-norpA-RNAi* line (JF01585) used in this study has positive validation data for knockdown efficacy, showing larval thermotaxis defects following nervous system knockdown (*Shen et al., 2011*). The second *UAS-norpA-RNAi* line (JF01713) does not have positive validation data per the RSVP database.

### Microarray analysis of mdIV gene expression

To confirm expression of *Gαq* and *norpA* in the mdIV neurons, an existing microarray dataset (ArrayExpress Accession # E-MTAB-3863) was analyzed. This dataset was created using the Affymetrix GeneChip *Drosophila* Genome 2.0 Array and contains data from four biological replicates, each consisting of 40–50 mdIV neuron cell bodies collected via laser capture microdissection (*Honjo et al., 2016*; *Mauthner et al., 2014*). Normalized detection values for *Gαq* and *norpA* transcripts were compared to those for the *trio, dTrpA1, painless,*

and *Gr28b* transcripts, all of which are known to be expressed in the mdIV neurons (*Iyer et al., 2012*; *Tracey et al., 2003*; *Xiang et al., 2010*; *Zhong et al., 2012*).

## Thermal nociception assays

Thermal nociception assays were conducted as previously described (*Tracey et al., 2003*). Wandering 3rd instar larvae were washed from vials into glass petri dishes with distilled water. A small amount (<10 mg) of dry baker's yeast was added to each petri dish to disrupt surface tension of the water, and enough water was removed from the dish to leave only a thin film of water covering the surface. Larvae were then stimulated along their lateral surface with a custom-built thermal probe consisting of a soldering iron with a tip filed into a ~6 mm chisel shape plugged into a Variac Variable Transformer (Part No. ST3PN1210B) (ISE, Inc., Cleveland, OH) to control probe temperature. The probe temperature was monitored using an IT-23 thermistor and BAT-12 digital thermometer (Physitemp, Clifton, NJ). The thermistor and thermometer provided temperature information to a tenth of a degree. During experiments, the probe tip temperature was monitored before and after application of the stimulus. Trials in which the probe temperature deviated from the desired temperature by ±0.5 °C were discarded during analysis. Behavioral responses were recorded at 30 frames per second using a digital video camera mounted on a dissecting microscope. Adobe Premier Pro was used to determine the time of probe contact with the larva and execution of NEL for each trial in order to calculate response latency. Larvae that did not respond to the probe within ten seconds were scored as 11 for all subsequent analysis. At least 40 larvae were tested for each genotype in each experiment. Experimenters were blinded to larval genotype during stimulation and scoring. The Wilcoxon Rank-Sum Test was used to determine statistical significance of latency differences between control and experimental genotypes. Bonferroni correction was used to correct the $\alpha$ value for multiple comparisons in experiments with more than one experimental group compared to the control.

## Mechanical nociception assays

Mechanical nociception assays were conducted as previously described (*Zhong, Hwang & Tracey, 2010*). Larvae were prepared in petri dishes as described for thermal nociception assays. Mechanical stimulation was delivered with a custom-built Von Frey filament consisting of a 10 mm length of 8 lb. test nylon fishing line (Stren Original Monofilament 8 lb. line, Part #1304152, Pure Fishing, Inc., Columbia, SC) affixed to a Pasteur pipette. This Von Frey filament delivers a force of ~50 millinewtons. Each larva was stimulated with the Von Frey filament and NEL was scored as a binary variable in order to calculate the proportion of larvae that responded to the stimulus. Each larva was tested three times, but only the first stimulus was used for subsequent analysis. At least 100 larvae were tested per genotype in each experiment except for the experiment shown in Fig. S1B, in which >50 larvae were tested per genotype. Experimenters were blinded to larval genotype during stimulation and scoring. The Chi-square test was used to test for statistically significant differences in proportion between control and experimental genotypes. Bonferroni correction was used to correct the $\alpha$ value for multiple comparisons in experiments with more than one experimental group compared to the control.

## Confocal microscopy

Wandering 3rd instar larvae were immobilized by circumferential ligation by a hair around segment A3. This manipulation paralyzed all body segments posterior to the ligation. Immobilized larvae were then mounted in glycerol between two glass coverslips and subjected to confocal microscopy using a Zeiss LSM 880 microscope with a 488 nm laser line. Tiled z-stacks were obtained in order to capture the full arborization of ddaC mdIV neurons from body segments posterior to the ligation. Neurons from the A8 body segments were not captured or analyzed, as these appeared to have qualitatively different arborization from more anterior segments. Dendrites were analyzed using the NeuronJ plugin for ImageJ (*Meijering et al., 2004*). In brief, dendrites were traced manually in order to measure the total length of individual neurons' arborizations and to count the total number of branches present. Experimenters were blind to larval genotype during tracing and analysis. Student's *t*-test was used to test for statistically significant differences between control and experimental conditions. Experimenters were blinded to genotype during this analysis to avoid bias in dendrite tracing.

## RESULTS

### *Gαq* is required for normal sensitivity to noxious thermal and mechanical stimuli

In order to test the role of Gαq signaling in larval nociception, we tested larvae with nociceptor-specific *Gαq* knockdown for defects in thermal and mechanical nociception. Before knocking down *Gαq* with RNAi, we confirmed that *Gαq* transcript is present in the nociceptors under normal conditions. We analyzed an existing Affymetrix microarray dataset created using mdIV cell bodies collected using laser capture microdissection(*Honjo et al., 2016*; *Mauthner et al., 2014*). We found that *Gαq* transcript was detected at a similar or higher level to *trio, dTrpA1, painless,* and *Gr28b* transcripts, all of which are known to be expressed in the mdIV neurons (Fig. S1) (*Iyer et al., 2012*; *Tracey et al., 2003*; *Xiang et al., 2010*; *Zhong et al., 2012*).

In our first behavioral experiment, we used the *ppk-GAL4; UAS-dicer2* line to drive nociceptor-specific expression of the GL01048 *UAS-Gαq-RNAi* transgene and tested for defects in thermal nociception. We found that *Gαq* RNAi larvae had significantly longer latency to respond to a 46 °C probe than controls with no RNAi transgene (Fig. 1A). *Gαq* RNAi larvae displayed a mean latency of 3.0 s, compared to a mean latency of 1.9 s for GAL4-only controls. During this experiment and all subsequent experiments, *para* knockdown larvae were used as a positive control and found to show nearly complete insensitivity to noxious thermal stimuli (Fig. 1A).

To confirm the apparent hyposensitive thermal nociception phenotype of *Gαq* knockdown larvae, we tested two additional *UAS-Gαq-RNAi* lines (carrying the JF02390 and JF02464 transgenes) for defects in thermal nociception (Fig. 1B). We found that larvae with *ppk-GAL4*-driven expression of the JF02390 RNAi transgene had a significantly longer latency to respond to a 46 °C probe than GAL4-only controls (2.2 versus 1.7 s). Larvae with *ppk-GAL4*-driven expression of the JF02464 RNAi transgene did not differ significantly

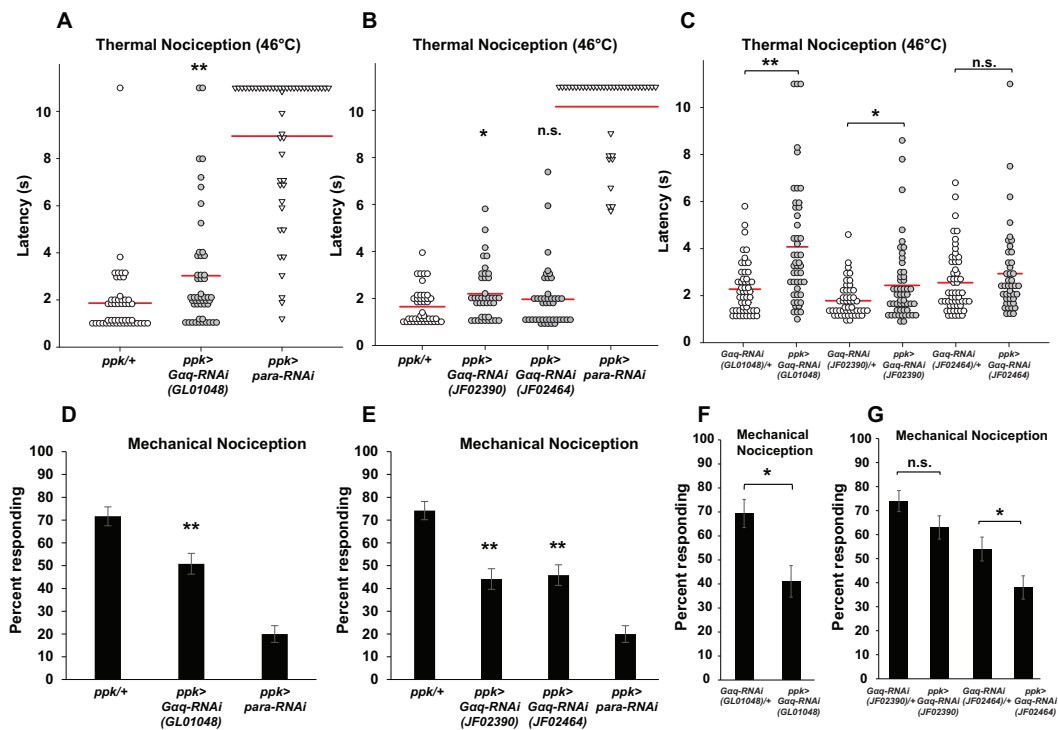

**Figure 1  Nociceptor-specific knockdown of $G\alpha q$ causes defects in thermal and mechanical nociception.** (A, B). Larvae with nociceptor-specific knockdown of $G\alpha q$ showed a significantly longer latency to respond to a noxious thermal stimulus (46 °C) than did GAL4-only controls. Larvae with nociceptor-specific knockdown of *para* showed severely impaired nociceptive responses and were used as a positive control. Response latencies of individual animals are plotted as points on the graph, while the mean for each genotype is indicated as a horizontal bar ($n \geq 40$ for all groups; $*p \leq 0.05$ by Wilcoxon Rank-Sum Test; $**p \leq 0.001$ by Wilcoxon Rank-Sum Test). (C) Larvae with nociceptor-specific knockdown of $G\alpha q$ showed a significantly longer latency to respond to a noxious thermal stimulus than did UAS-RNAi-only controls. ($n \geq 40$ for all groups; $*p \leq 0.05$ by Wilcoxon Rank-Sum Test; $**p \leq 0.001$ by Wilcoxon Rank-Sum Test). (D, E) A smaller proportion of larvae with nociceptor-specific knockdown of $G\alpha q$ exhibited nociceptive responses to a noxious mechanical stimulus than did GAL4-only control larvae. Larvae with nociceptor-specific knockdown of *para* showed a very low rate of nociceptive responses and were used as a positive control ($n = 120$ for all groups; $**p \leq 0.001$ by Chi-Square Test). Bars indicate the proportion of animals from each genotype that responded to the first application of the mechanical stimulus. Error bars indicate the standard error of the proportion. (F, G) A smaller proportion of larvae with nociceptor-specific knockdown of $G\alpha q$ exhibited nociceptive responses to a noxious mechanical stimulus than did UAS-RNAi-only control larvae. ($n > 50$ for all groups in *GL01048* graph and $n = 100$ for all other groups; $*p \leq 0.05$ by Chi-Square Test).

from GAL4-only control larvae in their response latency (1.9 versus 1.7 s). We also compared *ppk-GAL4*-driven expression of *UAS-Gαq-RNAi* to UAS-RNAi-only controls (Fig. 1C). We found that larvae with *ppk-GAL4*-driven expression of the GL01048 and JF02390 RNAi transgenes had significantly longer response latencies than their respective UAS-RNAi-only controls (4.1 versus 2.3 s for GL01048; 2.4 versus 1.8 s for JF02390). Larvae with *ppk-GAL4*-driven expression of the JF02464 RNAi transgene did not differ significantly from UAS-RNAi-only control larvae in their response latency (2.9 versus

2.6 s). Taken together, these results suggest that $G\alpha q$ knockdown produces a hyposensitive thermal nociception phenotype.

In order to determine whether the hyposensitive $G\alpha q$ knockdown phenotype is specific to thermal nociception or present for other nociceptive modalities, we also tested nociceptor-specific $G\alpha q$-*RNAi* for defects in mechanical nociception. In our first experiment, we used the *ppk-GAL4; UAS-dicer2* line to drive nociceptor-specific expression of the GL01048 *UAS-Gαq-RNAi* transgene and stimulated larvae with a 10 mm Von Frey filament to induce nociceptive responses. We found that larvae with nociceptor-specific $G\alpha q$ knockdown showed a significantly lower frequency of nociceptive responses compared to a control with no RNAi transgene (50.8% of larvae responding to first stimulus versus 71.7% of larvae responding to first stimulus) (Fig. 1D). To confirm this hyposensitive mechanical nociception phenotype, we also tested the responses of the JF02390 and JF02464 *UAS-Gαq-RNAi* lines to the same noxious mechanical stimulus. We found that each of these RNAi lines also produced a defective mechanical nociception phenotype (44.2% of JF02390 larvae and 45.8% of JF02464 larvae responding to the first stimulus versus 74.2% of GAL4-only control larvae responding to the first stimulus) (Fig. 1E). We also compared *ppk-GAL4*-driven expression of *UAS- Gαq-RNAi* to UAS-RNAi-only controls (Figs. 1F and 1G). We found that larvae with *ppk-GAL4*-driven expression of the GL01048 and JF02464 RNAi transgenes responded at a lower frequency than their respective UAS-RNAi-only controls (69.4% versus 41.1% for GL01048; 54% versus 38% for JF02464). Larvae with *ppk-GAL4*-driven expression of the JF02390 RNAi transgene did not differ significantly from UAS-RNAi-only control larvae in their response rate (74% versus 63% responding).

## Overexpression of *Gαq* results in hypersensitive thermal and mechanical nociception

Our observation that nociceptor-specific $G\alpha q$ knockdown results in hyposensitive thermal and mechanical nociception phenotypes suggests that Gαq signaling normally acts to increase the basal sensitivity of the mdIV neurons to noxious stimuli. To test this hypothesis, we overexpressed Gαq in the mdIV neurons by driving expression of a *UAS-Gαq* transgene with the *ppk-GAL4* driver and tested larval responses to noxious thermal and mechanical stimuli. We found that $G\alpha q$-overexpressing larvae responded significantly faster than UAS-only and GAL4-only control larvae to a 46 °C thermal stimulus (Fig. 2A). Overexpression larvae responded with an average latency of 2.2 s, while UAS-only and GAL4-only controls responded with latencies of 2.8 s. The latency for overexpression larvae was significantly shorter than both controls. These results are consistent with $G\alpha q$ overexpression causing hypersensitivity to noxious thermal stimuli.

To determine whether $G\alpha q$ overexpression also leads to increased sensitivity to mechanical stimuli, we tested overexpression larvae for mechanical nociception responses. We found that a significantly greater proportion of larvae with nociceptor-specific $G\alpha q$ overexpression responded to a noxious mechanical stimulus than did *UAS-Gαq*-only control larvae (84.0 percent responding to the first stimulus versus 65.3 percent responding to the first stimulus) (Fig. 2B). However, the difference in proportion of $G\alpha q$ overexpression larvae and GAL4-only control larvae that responded to the mechanical

**A**

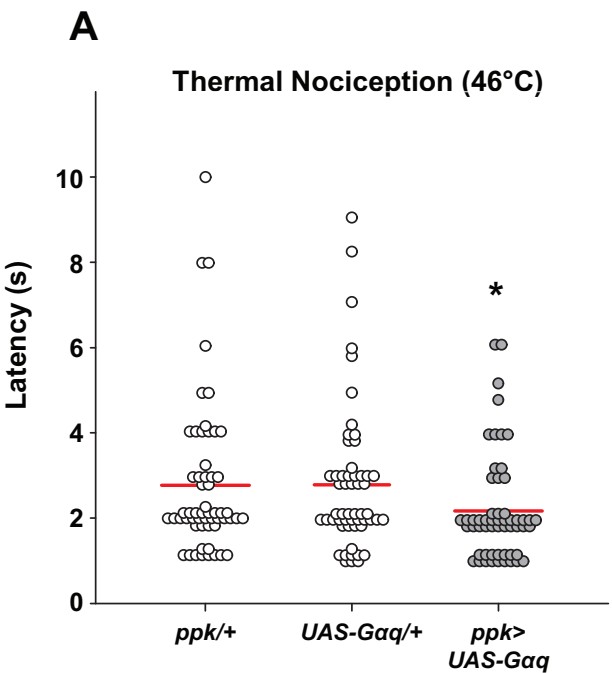

**B**

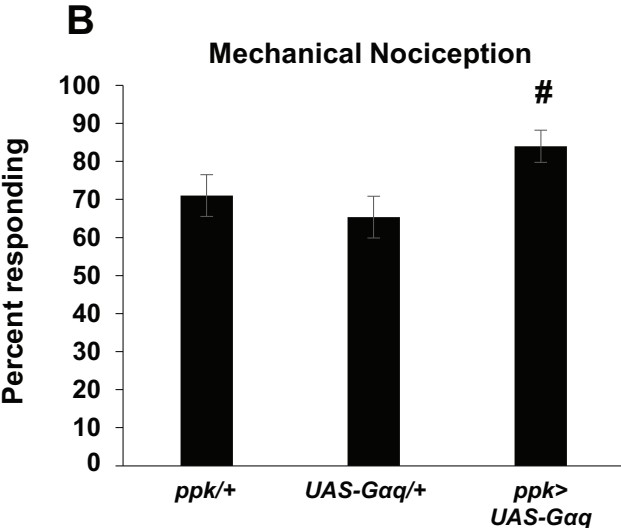

**Figure 2  Nociceptor-specific overexpression of *Gαq* causes hypersensitivity to noxious thermal and mechanical stimuli.** (A) Larvae with nociceptor-specific overexpression of *Gαq* showed a significantly shorter latency to respond to a noxious thermal stimulus (46 °C) than did GAL4-only or UAS-only control larvae. Response latencies of individual animals are plotted as points on the graph, while the mean for each genotype is indicated as a horizontal bar ($n \geq 50$ for all groups; *$p \leq 0.05$ by Wilcoxon Rank-Sum Test compared to both control genotypes). (B) A greater proportion of larvae with nociceptor-specific overexpression of *Gαq* responded to a noxious mechanical stimulus than did UAS-only controls ($n \geq 65$ for all groups; #$p \leq 0.05$ by Chi-Square Test compared to UAS-only control). Bars indicate the proportion of animals from each genotype that responded to the first application of the mechanical stimulus. Error bars indicate the standard error of the proportion.

stimulus was not significantly different (84.0 percent versus 71.0 percent responding to the first mechanical stimulus). These results only partially support an effect of *Gαq* overexpression on mechanical nociception, as the overexpression genotype is significantly different from only one of two control conditions.

### *NorpA* is required for normal sensitivity to noxious thermal and mechanical stimuli

One of the principal effectors of Gαq is PLCβ, and one of the major PLCβ enzymes in *Drosophila* neurons is encoded by the *norpA* gene. *norpA* transcripts are detected in larval nociceptors by microarray at similar levels to other known mdIV transcripts (Fig. S1). Given the thermal and mechanical nociception phenotypes that we have observed in nociceptor-specific *Gαq* knockdown and overexpression animals and previous studies detailing roles for NorpA in thermosensation, we hypothesized that NorpA signaling may also be required in the mdIV nociceptors for normal sensitivity to thermal and mechanical stimuli. To test this hypothesis, we used the *ppk-GAL4* driver to express *norpA* RNAi in the nociceptors and tested larvae for nociception defects. First, we crossed the *ppk-GAL; UAS-dicer2* driver line with the JF01713 and JF01585 *UAS-norpA-RNAi* lines and tested responses of larval progeny to a 46 °C thermal probe (Fig. 3A). We found that larvae expressing the JF01713 *UAS-norpA-RNAi* transgene possessed a significantly longer latency to the noxious thermal stimulus than the GAL4-only controls (3.8 s versus 1.7 s). However, larvae expressing the JF01585 *UAS-norpA-RNAi* transgene were indistinguishable from GAL4-only controls in their latency to respond to a thermal stimulus (1.7 s for both genotypes). We also compared *ppk-GAL4*-driven expression of *UAS-norpA-RNAi* to UAS-RNAi-only controls (Fig. 3B). We found that larvae expressing the JF01713 *UAS-norpA-RNAi* transgene possessed a significantly longer latency to the noxious thermal stimulus than UAS-RNAi-only controls (4.1 s versus 2.2 s). However, larvae expressing the JF01585 *UAS-norpA-RNAi* transgene were indistinguishable from UAS-RNAi-only controls in their latency to respond to a thermal stimulus (2.0 s versus 2.3 s). These results are consistent with a modest effect of *norpA* knockdown on thermal nociception.

Because the two *norpA* RNAi lines that we tested did not produce a consistent thermal nociception phenotype, we also tested a *norpA* mutant for thermal nociception defects. The *norpA*[36] mutant is a 28 bp deletion in the *norpA* gene that produces a frameshift in the *norpA* transcript and a truncated protein (*Pearn et al., 1996*). We found that the *norpA*[36] mutant responded to a noxious thermal stimulus with a significantly longer latency than *w*[1118] control larvae (4.9 s versus 2.5 s) (Fig. 3C). Thus the *norpA* loss-of-function phenotype is consistent with the phenotype produced by nociceptor-specific knockdown using the JF01713 *UAS-norpA-RNAi* transgene.

To determine whether *norpA* knockdown larvae also have defects in mechanical nociception, we crossed the *ppk-GAL; UAS-dicer2* driver line with the JF01713 and JF01585 *UAS-norpA-RNAi* lines and tested the larval progeny for responses to a 10 mm Von Frey filament. We found that a significantly smaller proportion of larvae expressing the JF01585 *UAS-norpA-RNAi* transgene responded to the noxious mechanical stimulus than GAL4-only control larvae (55 percent responding to the first stimulus versus 71.7

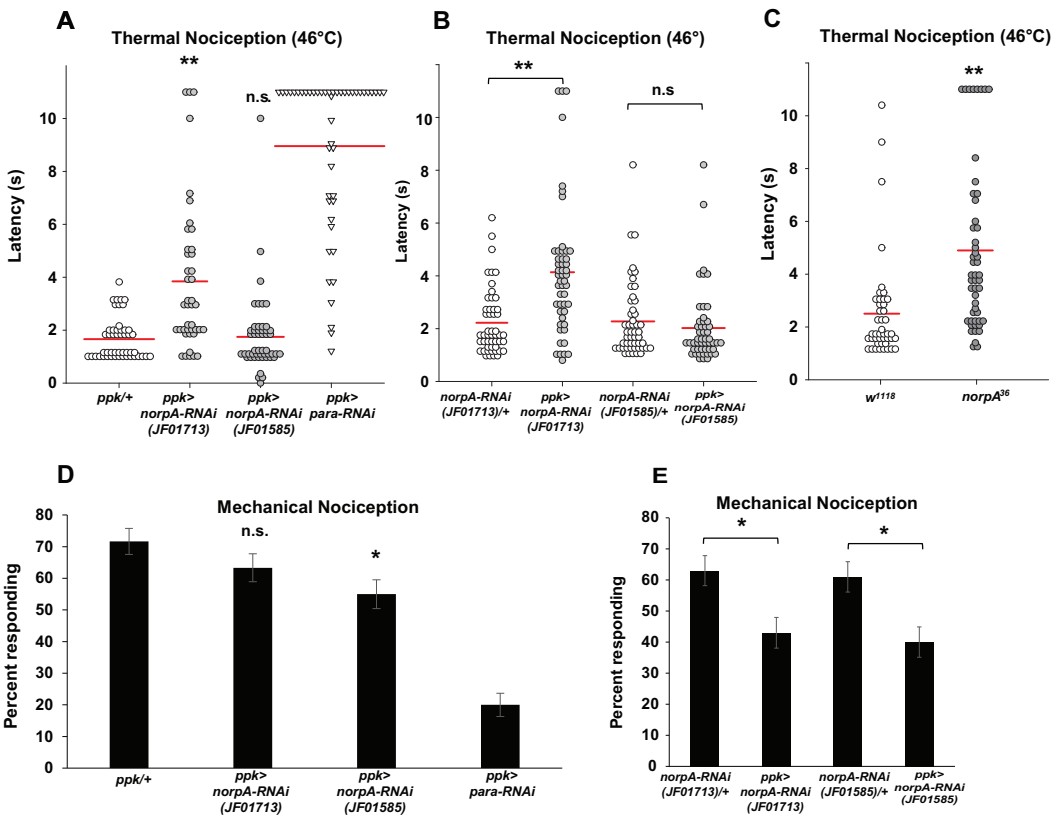

**Figure 3** **Loss of *norpA* function in the nociceptors causes defects in thermal and mechanical nociception.** (A) Larvae with nociceptor-specific knockdown of *norpA* using the JF01713 *UAS-norpA-RNAi* transgene respond to a noxious thermal stimulus (46 °C) with a significantly longer latency than do GAL4-only control larvae. Larvae with nociceptor-specific knockdown of *norpA* using the JF01585 *UAS-norpA-RNAi* transgene respond to noxious thermal stimuli with a mean latency that is not distinguishable from the control. Response latencies of individual animals are plotted as points on the graph, while the mean for each genotype is indicated as a horizontal bar ($n \geq 40$ for all groups; **$p \leq 0.001$ by Wilcoxon Rank-Sum Test). (B) Larvae with nociceptor-specific knockdown of *norpA* using the JF01713 *UAS-norpA-RNAi* transgene respond to a noxious thermal stimulus with a significantly longer latency than do UAS-RNAi-only control larvae. Larvae with nociceptor-specific knockdown of *norpA* using the JF01585 *UAS-norpA-RNAi* transgene respond to noxious thermal stimuli with a mean latency that is not distinguishable from the UAS-RNAi-only control. ($n \geq 40$ for all groups; **$p \leq 0.001$ by Wilcoxon Rank-Sum Test). (C) *norpA*[36] larvae respond to a noxious thermal stimulus with a significantly longer latency than do *w*[1118] control larvae. (D) A smaller proportion of larvae with nociceptor-specific knockdown of *norpA* using the JF01585 *UAS-norpA-RNAi* transgene exhibited nociceptive responses to a noxious mechanical stimulus than did GAL4-only control larvae. The proportion of larvae with knockdown of *norpA* using the JF01713 *UAS-norpA-RNAi* transgene that responded to a noxious mechanical stimulus was indistinguishable from that of GAL4-only control animals ($n = 120$ per group; *$p \leq 0.05$ by Chi-Square Test). Bars indicate the proportion of animals from each genotype that responded to the first application of the mechanical stimulus. Error bars indicate the standard error of the proportion. (E) A smaller proportion of larvae with nociceptor-specific knockdown of *norpA* using the JF01713 and JF01585 *UAS-norpA-RNAi* transgenes exhibited nociceptive responses to a noxious mechanical stimulus than did UAS-RNAi-only control larvae. ($n = 120$ per group; *$p \leq 0.05$ by chi-square test).

percent responding to the first stimulus) (Fig. 3D). However, the proportion of larvae expressing the JF01713 *UAS-norpA-RNAi* transgene that responded to the stimulus was not significantly different from GAL4-only controls (63.3 percent responding versus 71.7 percent responding). We also compared *ppk-GAL4*-driven expression of *UAS-norpA-RNAi* to UAS-RNAi-only controls (Fig. 3E). We found that larvae expressing the JF01713 and JF01585 *UAS-norpA-RNAi* transgenes both responded at significantly lower frequency than their respective UAS-RNAi-only controls (63% versus 43% for JF01713; 61% versus 40% for JF01585). These results are consistent with a mechanical nociception defect produced by *norpA* knockdown using either UAS-RNAi transgene.

### *Gαq* and *NorpA* are not required for nociceptor dendrite morphogenesis

Nociception defects in *Drosophila* larvae may arise from defects in mdIV development and morphogenesis. In order to determine whether manipulation of Gαq and NorpA signaling causes nociception defects via influence on mdIV dendritic arborization, we analyzed the arborizations of mdIV neurons in *Gαq* and *norpA* knockdown larvae. We used the *ppk-GAL4* driver to express mCD8::GFP and *Gαq* and *norpA* RNAi in the mdIV neurons and then imaged the dendritic arborizations of individual ddaC mdIV neurons in immobilized larvae (Figs. 4A–4C). We then counted the total number of dendrite branches and measured the total dendrite length of individual ddaC neurons in order to provide a quantitative measure of dendrite arborization. We found that *Gαq* and *norpA* knockdown (using the JF02390 and JF01713 RNAi lines respectively) did not significantly affect the total length or number of ddaC dendrites as compared to a no-RNAi control (Figs. 4D and 4E). These data suggest that *Gαq* and *norpA* knockdown does not produce gross morphological changes in the mdIV neurons.

## DISCUSSION

We have demonstrated that nociceptor-specific knockdown of *Gαq* knockdown causes *Drosophila* larvae to respond to noxious thermal stimuli with longer response latencies and to noxious mechanical stimuli with reduced frequency. These results suggest a modest role for Gαq in positively regulating nociceptor sensitivity. This interpretation is supported by our observation that *Gαq* overexpression in the nociceptors causes faster and more frequent responses to thermal and mechanical stimuli respectively. NorpA is an effector of Gαq in phototransduction and thermotransduction in *Drosophila* (*Lee et al., 1994*; *Running Deer, Hurley & Yarfitz, 1995*; *Shen et al., 2011*). Given our observation that *Gαq* and *norpA* knockdown larvae share similar defective nociception defects, we hypothesize that NorpA also acts as a Gαq effector in larval nociceptor neurons to regulate sensitivity to noxious thermal and mechanical stimuli. This hypothesis could be formally tested by epistasis experiments using Gαq gain-of-function flies. Loss of Gαq and NorpA function in the nociceptors does not result in gross defects in dendrite development or arborization, suggesting that nociception defects do not arise from defects in multidendritic neuron development. Taken together, these results demonstrate that larval nociception is a

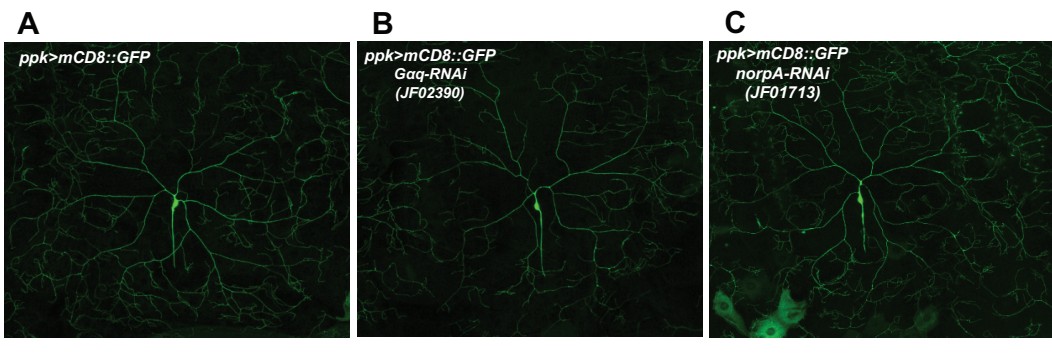

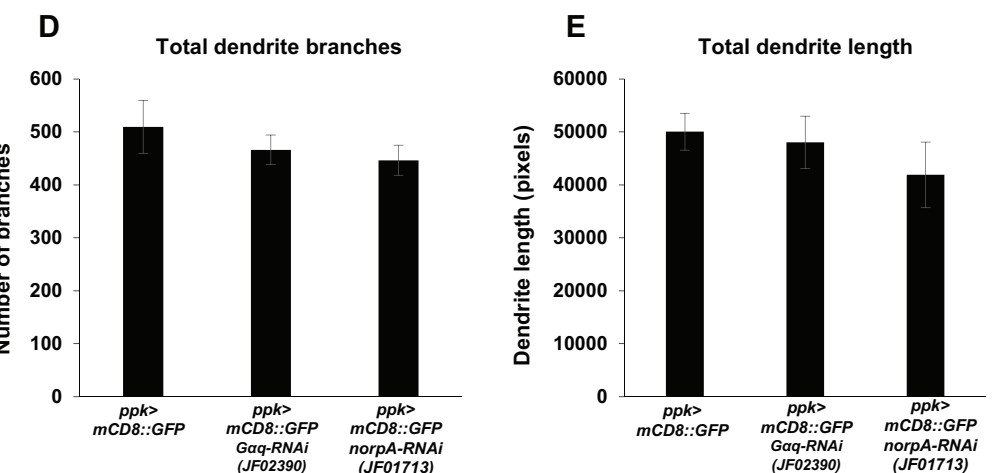

**Figure 4** **Nociceptor-specific knockdown of *Gαq* and norpA does not cause changes in mdIV neuron dendrite length or branch number.** (A, B, C) Confocal micrographs displaying the dendritic arborization of ddaC mdIV neurons expressing mCD8::GFP and *Gαq* or *norpA* RNAi. (D) The total numbers of dendrite branches in ddaC dendrites with *Gαq* or *norpA* knockdown were not significantly different (by Student's *t*-test) from those of no-RNAi control ddaC neurons. (E) The total lengths of ddaC dendrites with *Gαq* or *norpA* knockdown were not significantly different (by Student's *t*-test) from those of no-RNAi control ddaC neurons. Bars indicate mean total dendrite length, while error bars indicate standard error of the mean ($n = 6$ neurons for each group).

promising experimental paradigm for further study of the role of Gαq and NorpA signaling in modulation of nociceptor sensitivity.

The proposed roles for Gαq and NorpA signaling in regulating nociceptor sensitivity are largely, but not perfectly, supported by the experimental data. Two of three *UAS-Gαq-RNAi* transgenes and one of two *UAS-norpA-RNAi* transgenes produce thermal nociception defects when expressed in the larval nociceptors. Roles for Gαq and NorpA in positive regulation of thermal nociception are further supported, however, by the observations that nociceptor-specific Gαq overexpression produces a hypersensitive thermal nociception phenotype and that a *norpA* loss-of-function mutant shows a hyposensitive thermal nociception phenotype, as would be predicted by the cell-specific RNAi data. Roles for Gαq and NorpA signaling in mechanical nociception are strongly supported by two

of three *UAS-Gαq-RNAi* lines and one of two *UAS-norpA-RNAi* lines. The remaining *UAS-Gαq-RNAi* line and *UAS-norpA-RNAi* line provide only partial support for roles in mechanical nociception, as the knockdown phenotype produced by each is significantly different from only one of the GAL4-only or UAS-RNAi-only controls. We also note that the effects of *Gαq* and *norpA* knockdown are relatively modest. These observations might be explained by the vast signal amplification potential of Gαq and NorpA signaling (*Hardie et al., 2002*). It is possible that modest residual levels of Gαq and NorpA signaling following knockdown could support unexpectedly high levels of second messenger production. In this scenario, only the strongest UAS-RNAi lines would be expected to produce a significant change in nociceptor sensitivity. This hypothesis could be investigated by further analysis of *Gαq* and *norpA* mutants as well as antibody staining experiments to quantify the effects of knockdown on Gαq and NorpA protein levels.

While we have hypothesized that NorpA is the major effector of Gαq in larval nociceptors, it is also possible that Gαq signals through other effectors aside from NorpA in the mdIV neurons. In some systems, the Trio rhoGEF is directly activated by Gαq signaling (*Lutz et al., 2005*; *Lutz et al., 2007*; *Rojas et al., 2007*; *Williams et al., 2007*), and recent studies have demonstrated roles for Trio in regulating mdIV morphogenesis (*Brown et al., 2017*; *Iyer et al., 2012*). However, Trio loss-of-function larvae do not display mechanical nociception defects, suggesting that Trio may not be an effector of Gαq for the modulation of nociceptor sensitivity (*Brown et al., 2017*). However, more targeted epistasis experiments may be needed to formally investigate this possibility.

The downstream effectors of Gαq and NorpA signaling in the mdIV neurons remain to be determined. dTRPA1 ion channel function is required in larval nociceptors for thermal and mechanical nociception (*Neely et al., 2011*; *Zhong et al., 2012*), and dTRPA1 is known to be activated downstream of NorpA in *Drosophila* chemosensory neurons and to support thermotaxis behavior (*Kim et al., 2010*; *Kwon et al., 2010*; *Kwon et al., 2008*; *Shen et al., 2011*). Thus it is reasonable to hypothesize that dTRPA1 in the mdIV neurons is activated less effectively in the absence of Gαq and NorpA signaling, leading to decreased nociceptor sensitivity. One possible mechanism for the activation of dTRPA1 downstream of NorpA is that dTRPA1 may be activated by depletion of PIP2. With this in mind, it is important to note that PIP2 affects the activity of many types of ion channels (for review see *Hille et al., 2015*). PIP2 hydrolysis by NorpA may regulate the function of any number of ion channels that control nociceptor sensitivity, including voltage-gated calcium channels and small-conductance potassium channels (*Neely et al., 2010*; *Walcott et al., 2017*). We also cannot rule out that the generation of IP3 and DAG second messengers by PIP2 hydrolysis is the principal mechanism by which NorpA regulates nociceptor sensitivity, as these mechanisms are well known to mediate store-operated calcium release, activation of protein kinases, and regulation of the neurotransmitter release machinery.

The observed role of Gαq and NorpA in nociception suggests the existence of a GPCR signaling mechanism that activates this signaling pathway under basal conditions (i.e., in the absence of tissue damage or sensitization). The identity of this GPCR or these GPCRs remains to be discovered. Activation of sNPF receptors on the mdIV neurons facilitates mechanical nociception, presumably via a heterotrimeric G protein signaling mechanism

(*Hu et al., 2017*). It is possible that loss of Gαq or NorpA signaling prevents this sNPF facilitation of mechanical nociception, thus producing a defective mechanical nociception phenotype. However, signaling through sNPF receptors was found to facilitate mechanical nociception specifically, while our results suggest that Gαq and NorpA signaling facilitates both thermal and mechanical nociception. Thus, we may hypothesize that additional GPCRs exist to facilitate thermal nociception through heterotrimeric G protein signaling under basal conditions. The identities of these putative receptors and their ligands are a promising subject of further study.

## CONCLUSIONS

Our studies demonstrate that Gαq and PLCβ signaling acts in the nociceptors of *Drosophila* larvae to support wild-type sensitivity to noxious thermal and mechanical stimuli. This conclusion is supported by the fact that nociceptor-specific RNAi knockdown of either *Gαq* or *norpA* produces hyposensitive thermal and mechanical nociception phenotypes. Additionally, overexpression of *Gαq* causes thermal and mechanical hypersensitivity. The behavioral phenotypes observed following RNAi knockdown of *Gαq* or *norpA* are unlikely to arise from deficits in sensory neuron morphogenesis, as knockdown animals were found to have dendrites with similar length and branching to wild-type animals.

## ACKNOWLEDGEMENTS

Stocks obtained from the Bloomington Drosophila Stock Center were used in this study. We thank the TRiP at Harvard Medical School for providing transgenic RNAi fly stocks used in this study. We would like to thank Dr. Guichuan Hou and the William C. and Ruth Ann Dewel—College of Arts & Sciences (Dewel-CAS) Microscopy Facility for confocal microscopy support. We would also like to thank Dr. Dan Tracey who provided fly stocks used in this study and all of the members of the Bellemer lab who provided scientific and technical support.

### Funding

The Dewel-CAS confocal microscope is supported by the National Science Foundation (NSF-MRI 1625779). The funders had no role in study design, data collection and analysis, decision to publish, or preparation of the manuscript.

### Grant Disclosures

The following grant information was disclosed by the authors:
National Science Foundation: NSF-MRI 1625779.

### Competing Interests

The authors declare there are no competing interests.

## Author Contributions

- Joshua A. Herman conceived and designed the experiments, performed the experiments, analyzed the data, prepared figures and/or tables, approved the final draft.
- Adam B. Willits performed the experiments, analyzed the data, prepared figures and/or tables, approved the final draft.
- Andrew Bellemer conceived and designed the experiments, performed the experiments, analyzed the data, prepared figures and/or tables, authored or reviewed drafts of the paper, approved the final draft.

## Data Availability

The raw data are provided in the Supplemental Files.

## Supplemental Information

Supplemental information for this article can be found online at http://dx.doi.org/10.7717/peerj.5632#supplemental-information.

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
