# Peer review of "Gαq and Phospholipase Cβ signaling regulate nociceptor sensitivity in Drosophila melanogaster larvae"

_PeerJ, doi:10.7717/peerj.5632_

## Round 0.1 · original submission · Major Revisions

Please address the comments made by reviewer#2 and 3.

[# Staff Note: Reviewer 1 declared a potential Conflict of Interest. The Editor was aware of this when making their decision #]

·

Basic reporting

The article is clear and unambiguous. Sufficient background and context are provided. The figures are professional and the article is self contained with relevant results to hypotheses.

Experimental design

The research is original within the scope of the journal.

Research question is well defined.

The investigation is rigorous.

Methods are described with sufficient detail and information to replicate.

Validity of the findings

The study presents mainly positive data and it also includes one instance of negative data where a NorpA RNAi line does not have a phenotype.

The data are very nicely presented with all the points of data represented in the plots.

Conclusions are well stated.

There is little to no speculation.

Additional comments

The vast amplification that is possible with G-protein coupled signaling cascades may make RNAi approaches relatively ineffective. Tissue specific Cas9 expression will soon be available for targeting the cIVda neurons. A targeted CRISPR KO might result in a more severe phenotype than has been observed in the study with an RNAi approach.

Reviewer 2 ·

Basic reporting

Throughout, by “insensitive”, the authors mean “hyposensitive”? To this reviewer, “insensitive” means lack of sensitivity, not decreased sensitivity.

188-195, these UAS-RNAi lines are driven by ppk-Gal4, correct?

The Figure 1 legend should describe the temperature of the thermal probe.

213-215, remove interpretation of results to the Discussion section

238, 318 I think instead of “principle”, the authors mean “principal”

343 “branch” should be “branching”?

251-252, remove interpretation to Discussion

286-287 “NorpA is likely an effector of Gαq in the nociceptors, given the defective nociception phenotypes observed in norpA knockdown larvae.” Does this result really imply any causal relationship? Could be coincidental? An epistasis experiment would help with this.

Experimental design

The operator of the thermal probe should be blinded to the genotype. If this was indeed the case, or there is some other method of reducing the possibility of unintentional operator bias, those details should be included in the description of the method.

What is the accuracy of the temperature-control system of the thermal probe? How much variability exists in the temperature at the contact point under experimental conditions?

For von Frey probes, it is better to refer not to length (since diameter is also a factor) but to the force delivered, expressed in millinewtons…

Given the modest phenotypic effects observed and sometimes equivocal relationship to IR genotype, does N=40 give sufficient power to resolve? Has a power analysis been conducted?

Validity of the findings

This reviewer is concerned about the apparently inconsistent application of appropriate controls. In some experiments, such as Fig 2, comparison of Gal4/UAS experimental genotypes are appropriately compared with both Gal4-only and UAS-only controls. In contrast, Fig 1 and Fig 3 present just one negative control: Gal4-only. (Fig 4 appropriately depicts only one control: the Gal4-only control.) No rationale is given for the omission of missing controls. Were the unpresented controls not analyzed? In this reviewer’s opinion, both controls should be analyzed for experiments like those in Figures 1-3. If only one is presented graphically, the results of the other should be available in a supplement or at least described in the text. Comparisons in which the phenotype of the experimental genotype is not significantly different from that of both UAS-only and Gal4-only controls are not on the firmest footing. These control for important effects like leakiness of UAS-RNAi constructs and phenotypes of Gal4 insertions themselves.

Figure 1: Modest hyposensitivity observed in Gaq under-expression studies of thermonociception, more convincing hyposensitivity observed in mechanonociception compared to Gal4-only. However UAS-only controls are not presented. The results depicted in Fig 1A separate from B, and C separate from D, the reason for this separation is not obvious. If UAS-only controls were analyzed and presented, there would be a separate UAS-only control genotype required for each UAS-RNAi line used. So without comparison to these controls, these results are not as convincing as they could be.

Figure 2: Modest hypersensitivity observed in Gaq over-expression studies of both thermo- and mechanonociception. The figure legend for 2A states significantly different compared with both controls, and both controls are depicted, so the claim is fully supported here. For 2B, both controls are presented, but experimental shows difference only to UAS-only. So while the control structure of this experiment is satisfactory, because the experimental shows difference to only one of the controls, the mechanonociception results are not as convincing as they could be.

Figure 3: For norpA, one RNAi line JF01713 showed modest decrement in thermosensitivity, significantly different from Gal4-only control. However the UAS-only control is not shown or described. But the RNAi line JF01585 shows no differences from Gal4-only control, UAS-only control not shown or described. For norpA RNAi and mechanonociception, RNAi line JF01585 showed hyposensitivity compared to Gal4-only, RNAi line JF01713 showed no differences compared to Gal4-only. So that is the opposite response from these lines, compared to thermonociception. So these results put the conclusions in doubt.

Figure 4: Morphometry of ddaC cells. These are the most dorsal mdIV cells, presumably the cells most stimulated by mechanical stimuli, but thermal stimuli were presented laterally. The controls presented are complete and appropriate. These results are not much in doubt, but they are negative results that while they help to interpret, do not directly test the major hypotheses.

This reviewer is concerned that no evidence is presented to confirm the effectiveness of the cell-specific suppression or overexpression of the targets, or other evidence of their normal expression in these neurons. For example, validated antibodies could be employed for immunohistological analysis of wild-type and manipulated tissues. Perhaps validated MiMIC lines are available that might indicate the normal expression patterns of the targets.

Additional comments

The authors are to be commended for clear writing style. The experimental design is to be commended for combined use of over-expression tools and multiple non-overlapping IR transgenes to produce cell specific under-expression. Analysis of nociceptor morphology is excellent and important for interpretation of any changes in sensitivity phenotypes. Excellent coverage of the state of the field and context for the experiments. Record-keeping is excellent.

Reviewer 3 ·

Basic reporting

Pass.

Experimental design

Pass.

Validity of the findings

Fail.

The data is not robust and therefore, weakens the conclusion.

Additional comments

In this manuscript, the authors characterize the role of Gq/norpA signaling on nociception in Drosophila larvae. It is an interesting study but I have the following concerns:

1) The effect of different Gq knockdown lines on thermal nociception is different (Fig 1). Similar inconsistency is seen with different norpA knockdown lines in Fig 3. It is important to show the knockdown efficiency of each line to check whether the amounts of Gaq and norpA transcripts dictate the outcome.

2) The latency after thermal stimulation for the ppk/+ line in Fig 1 are 1.9 and 1.7 seconds while in Fig 2 it is 2.8 seconds. Although this kind of variability is not uncommon, the Gq knockdown lines show the latency of 3.0 and 2.2 in Fig 1 whereas Gq overexpression line has a latency of 2.2. This shows that the Gq gain- and loss-of-function conditions have very weak effects on thermal nociception and this considerably weakens the premise of the manuscript.

3) Since the phenotypic effect of Gaq/norpA modulation is very weak, the authors should show how they affect ion influx in these cells in response to thermal stimuli.

---

## Round 0.2 · Minor Revisions

Please address reviewer #2's minor comments before final decision.

Reviewer 2 ·

Basic reporting

There are still instances of interpretations this reviewer previously failed to point out in the Results section (eg lines 217-219). This review is not even very comfortable with the Results headings that state conclusions. Results is only facts and little/no interpretation. The Discussion section should refer back to the Results and that is where you draw your conclusions.

About controls, great that all negative controls are presented, but some figures contain both Gal4-only and UAS-only and others in which the UAS-only is contained in Supplemental Figures 1,2. Both negative controls should be placed alongside the experimental in each figure, so that readers can conveniently compare without having to flip back and forth between two separate figures. Dispense with the Supplementary Figures, all that data is more clearly represented in the main body.

Where the authors have added description of comparison to UAS-only controls (for example Line 283, 285, 317 etc), in the previously-written text specify that you are describing comparison to Gal4-only controls. In other words, in each sentence identify which controls you are describing.

Still not sure that Figure 1 A/B and C/D need to be separated. This arrangement may mean something to the authors, but readers may miss that point and be confused about why they are presented separately. The choice is the authors'/editors', but all these results focus on the same hypothesis.

Rather than saying in the Methods that these "lines were validated by the database", instead say that some but not all of the lines have been previously validated as described in the database, before going on to describe that validation, as was done.

This reviewer suggests that briefly including this analysis would be helpful: "We also agree that data about normal expression of NorpA and Gαq in the mdIV would be valuable for the interpretation of our studies. We have used a publically available microarray dataset (ArrayExpress Accession # E-MTAB-3863) to show that NorpA and Gαq transcripts are detected in the mdIV neurons at levels comparable to transcripts of known mdIV genes (trio, painless, dTrpA1, Gr28b). We have omitted this analysis from the revised manuscript, but can include it if needed."

The authors provided explanation of the their thermal probe temperature verification, this should be included in Methods: "The thermistor and thermometer used to measure temperature at the probe provides temperature information to a tenth of a degree. During experiments, the probe tip temperature is monitored before and after application of the stimulus. Trials in which the probe temperature deviates from the desired temperature by +/- 0.5°C are discarded during analysis."

Experimental design

pass

Validity of the findings

pass

Additional comments

Excellent revision. My remaining concerns focus on presentation.

Reviewer 3 ·

Basic reporting

Same as the first review (Acceptable).

Experimental design

Same as the first review (Acceptable).

Validity of the findings

The authors have addressed my concerns in their revised manuscript.

---

## Round 0.3 · accepted · Accept

The revised manuscript is acceptable in its present form.

#